# Probabilistic Grid System for Indoor Mobile Localization Using Multi-Power Bluetooth Beacon Emulator

**DOI:** 10.3390/s25185635

**Published:** 2025-09-10

**Authors:** Barbara Morawska, Piotr Lipiński, Krzysztof Lichy, Marcin Tomasz Leplawy

**Affiliations:** Institute of Information Technology, Lodz University of Technology, Al. Politechniki 8, 93-590 Lodz, Poland; barbara.morawska@dokt.p.lodz.pl (B.M.); krzysztof.lichy@p.lodz.pl (K.L.); marcin.leplawy@p.lodz.pl (M.T.L.)

**Keywords:** indoor localization, Bluetooth low energy, signal coverage analysis

## Abstract

**Highlights:**

**What are the main findings?**
We introduce a novel BLE-based approach for indoor mobile device localization that uses spatial signal coverage from multi-power Bluetooth emulators instead of RSSI.The approach improves localization accuracy by generating a probability grid based on receiver range data; static tests achieved 1.83 m average error, and dynamic tests showed 70% of samples with 3.6 m error.

**What is the implication of the main finding?**
The proposed method enables more reliable real-world tracking of mobile devices compared to traditional RSSI-based approaches.The system is being implemented at trade fairs to track attendee movement and optimize stand distribution.

**Abstract:**

Despite extensive research, indoor localization techniques remain an open problem, with Bluetooth Low Energy (BLE) continuing to be a dominant technology even in the presence of ultrawideband and Bluetooth 5.1. This study proposes a novel approach for indoor mobile device localization using BLE. Unlike traditional methods relying on the Received Signal Strength Indicator (RSSI), this technique employs spatial signal coverage analysis from multi-power Bluetooth emulators, with data collected by an array of receivers. These coverage patterns form a probability grid, which is processed to accurately determine the mobile device’s location. The method accounts for the intrinsic properties of antennas and the operational ranges of multiple beacon emulators, thereby enhancing localization precision. By utilizing receiver range data rather than RSSI, localization outcomes demonstrate greater consistency. Static measurements show an average error of 1.83 m, a median error of 1.73 m, and a mode error of 2.35 m. In dynamic settings, a moving robot exhibited a measurement error of 3.6 m for 70% of samples and 4.6 m for 94% of samples. This solution is currently being implemented to track attendees at trade fairs, providing metrics to inform stand rental pricing and insights for optimizing stand distribution to encourage visitor exploration.

## 1. Introduction

Bluetooth beacons are devices based on Bluetooth 4.0 Low Energy (BLE) technology [1]. They are characterized by their compact dimensions, lightweight construction, economical price, and minimal power consumption, enabling operational longevity of up to several years without necessitating battery replacement. Furthermore, mobile phones can function as beacons with negligible impact on battery life. Such attributes attracted its application among others to localization in Global Navigation Satellite System-denied areas, which can be applied in many practical scenarios, such as, for example, interactive sightseeing described in [2], which allows museum visitors to familiarize themselves with details about an exhibition in an interactive and personalized way, thereby increasing its educational value. Another example is directional intelligent management by analyzing real-time crowd flow information in shopping malls, as discussed in [3]. Bluetooth beacons are also fundamental components of Smart Parking Systems [4], where they are employed to guide drivers to available parking spaces. Additionally, they are utilized in airports [5] for staff localization and in hospitals [6] to track patients and ensure their safety. Furthermore, during the COVID-19 pandemic, Bluetooth beacons proved valuable for detecting contacts with potentially infected individuals [7] and for implementing geofencing systems [8].

Bluetooth-based localization can be used in cases where satellite systems like GPS, GLONASS, QZSS, or Beidou do not meet accuracy expectations. However, developers usually implement solutions like that with beacons based on the principle of proximity, as seen in [2]. A visitor (transmitter) has to be in the range of the receiver, and then an audio track about a piece of art starts playing. The accuracy of such localization depends on the range of the transmitter, but with a small range, it can still be tens of square meters (see Figure 1b). That is why it makes sense to use an extra parameter to improve measurement error. The most common way to do that is a Received Signal Strength Indicator (RSSI), used as either a radio fingerprint or power loss [9,10,11]. The advantage of this solution is that in the simplest way, collecting RSSI is possible on almost every Bluetooth device, like smartphones, embedded devices, personal computers, etc. There are also more complex techniques that resort to finer-grained wireless channel measurement than just RSSI. Different from RSSI, as the MAC layer superimposition of multi-path signals with fast-changing phases, the PHY layer power feature, channel response, is able to discriminate multi-path characteristics [12]. However, devices to measure such parameters are expensive and difficult to access. Moreover, in 2019, a localization standard was released—Bluetooth 5.1 Direction Finding [13]—which enabled the use of the Angle of Arrival and Angle of Departure methods [14]. However, because of high device costs and low accuracy, it was not welcomed warmly in contrast to the same adaptation in similar radio technologies like ultrawideband, which is used, for example, in Apple AirTags. There are also less common graph methods, which are based on matching the measured sample to the location where the smallest error is made [15]. Additionally, there are graphical methods, which are approaches described in this article and also used in [16].

The aforementioned localization methods may be significantly improved through the application of data fusion techniques, such as the Kalman Filter (KF) [10,17], the Extended Kalman filter [18], the Unscented Kalman Filter [19], the Particle Filter [4], neural networks [20], and classification methods [21] (e.g., KNN—k-nearest neighbors [22,23]). However, the application of KF, EKF, and UKF necessitates that the signal errors exhibit a Gaussian distribution with a zero mean, conditions that are frequently unmet in indoor environments when employing radio frequency-based measurement techniques. Conversely, the implementation of PF requires precise knowledge of the forcing function, which is often challenging to ascertain in the context of mobile device localization. Moreover, methodologies that rely on neural networks and classification algorithms are highly sensitive to environmental changes and variations in the surroundings where measurements are conducted.

Generally, the problem with all methods based solely on RSSI as a value determining distance is that they suffer from dramatic performance degradation in complex situations due to multi-path propagation, reflections, signal attenuation, and temporal dynamics, which results in low localization accuracy and high measurement variations. The common solution is to pack the room with many beacons operating on low power so the results are more accurate and stable.

According to the above-mentioned literature, when the distance between beacons is 2 to 4 m, we can attain an accuracy of 1 to 2 m. Nevertheless, practical considerations, such as economic constraints and bandwidth limitations, often necessitate the reduction of deployed devices.

A pertinent industry example involves developing an indoor localization system for a large trade fair hall exceeding 100,000 m^2^. The objective was to determine optimal pricing and distribution of exhibition stands to maximize revenue, necessitating the tracking and recording of visitor movements and the frequency of stand visits.

The challenge was to design a large-scale, distributed sensor network capable of comprehensive visitor data collection while remaining cost-effective. BLE technology was selected as the primary technology due to its capability to track every visitor via their mobile phones and a custom application. This application provides a map of the trade fair, tickets, and event details and tracks user movements upon obtaining their consent. Additionally, equipping each exhibition stand with one or more BLE beacons proved feasible and cost-effective.

The main contribution of this paper is the introduction of a novel, cost-effective, BLE-based approach for indoor mobile device localization. Unlike traditional methods that rely on Received Signal Strength Indicator (RSSI), the proposed method leverages spatial signal coverage generated by multi-power Bluetooth emulators. By constructing a probability grid derived from receiver range data, the approach significantly enhances localization accuracy. Experimental validation demonstrated its effectiveness by outperforming similar solutions. Beyond the experimental setup, the system is currently being deployed at trade fairs to monitor attendee movement and support the optimization of stand distribution.

The system described herein effectively addresses the challenge by employing BLE technology to monitor visitor behavior.

## 2. Related Work

A comparable approach to ours is described in the article titled “RSSI-based Bluetooth Indoor Localization” [16]. It introduces two algorithms for indoor localization using Bluetooth Low Energy (BLE): Low-precision Indoor Localization (LIL) and the improved High-precision Indoor Localization (HIL). Both algorithms represent potential receiver coverage areas as pixel grids and combine these maps using logical operations. HIL refines LIL by dividing RSSI values into subgroups to further narrow the estimated transmitter area. The evaluation was performed in simulation, with four receivers placed at the corners of a 7 × 7 m area. HIL localized all measurements within areas no larger than 18 m^2^, whereas LIL produced larger localization regions. While conceptually similar in its use of RSSI-derived spatial masks, that work remains simulation-based; our system operates on real measurements and incorporates several practical enhancements that improve accuracy and robustness. Another relevant approach is presented in [15], which does not rely on RSSI or other signal strength indicators. Instead, it uses only the visibility of Bluetooth beacons (a simple true-or-false test of signal reception). However, this graph-based localization algorithm has not been tested in a real-world environment. Some methods are based on direction-finding techniques [24]. However, angle-of-arrival measurements require specialized hardware, in contrast to our approach. There are also methods that use Multi-Carrier Phase Difference [25], which are noteworthy for combining BLE ranging with IMU data through a particle filter. While effective, these methods also require specialized hardware, unlike our solution. Fingerprinting-based methods, such as those in [26] and LSTM-powered fingerprinting approaches [27], can achieve significantly higher accuracy than our system. However, the key difference is that these methods require prior environment scanning before measurements can be taken. In contrast, our system operates without any pre-deployment calibration—sensors can be used immediately after installation. Our contribution distinguishes itself by proposing a practical, hardware-agnostic BLE localization system capable of operating in real-world environments without the need for prior calibration. By integrating a grid-based spatial masking approach with noise-tolerant processing of live RSSI data, our method achieves reliable position estimation using only standard BLE devices. This allows for immediate deployment in dynamic environments where calibration, specialized hardware, or pre-scanning are impractical or resource-intensive.

## 3. Problem in Bluetooth Beacon Indoor Localization

A significant part of indoor systems employing Bluetooth beacons relies on RSSI. In theory, RSSI values received at the receiver can be converted into transmitter–receiver distances through a mathematical model based on Bessel functions [28]. However, practical implementation faces challenges stemming from the equation’s reliance on numerous environmental variables. Operating such a system indoors, where elements like windows, walls, doors, furniture composed of diverse materials, and human movement abound exist, poses difficulty in meeting anticipated performance levels due to the complexity of variables involved. Figure 2 depicts the relationship between received signal strength and the distance separating the transmitter and receiver [29]. The black, solid curve represents the mathematical model based on Bessel functions between these values. The black dots represent signal strength measurements taken at different distances. The dashed lines show the standard deviation from the solid curve. For instance, an RSSI of −75 dBm corresponds to distances ranging from 50 to 100 m. The RSSI value is subject to significant temporal fluctuations. Figure 3 depicts sample measurements performed in our laboratory that demonstrate RSSI value changes over time when both the transmitter and receiver are stationary.

The above examples demonstrate that localization based solely on converting RSSI to distance is not effective in real conditions. Therefore, applying well-known position determination methods like multi-lateration with Bluetooth beacons will not yield reliable results.

More commonly assumed but often overlooked aspects of localization using Bluetooth beacons are the assumption of an isotropic antenna pattern and the assumption that two identical devices placed in different locations have exactly the same error characteristics [22]. These simplifications also apply to Figure 1a,b.

However, in practical applications, these assumptions seldom hold true. While real antenna pattern shapes may be outlined in device documentation, success is not guaranteed. Placing the transmitter near dense walls or metallic surfaces can significantly distort the antenna pattern’s shape. Similarly, seemingly minor factors such as varying device orientation or enclosure can also produce similar effects. This phenomenon is demonstrated in Figure 4, where the antenna pattern is altered due to reflection. In Figure 4a, despite the receiver (R) being at the same distance from the transmitter (T) as in Figure 1b, it falls out of range. In another scenario (Figure 4b), where R is further from T than in Figure 1a, it remains within the transmitter’s range.

## 4. Algorithm

In the preceding section, a discussion ensued regarding the constraints and simplifications inherent in Bluetooth-based localization methodologies, potentially resulting in notable discrepancies. In this section, we present an algorithm for Bluetooth-based localization that is robust against such discrepancies.

In contrast to the BLE systems delineated in the introduction section, our system employs a role reversal of the receiver and transmitter. Specifically, within our framework, the mobile phone assumes the function of the beacon by broadcasting an Eddystone UUID beacon signal. Concurrently, the application installed on stationary receiver devices situated in predetermined localization coordinates remains stationary and intercepts signals emitted by the mobile phone. The UUID serves not only as the beacon identifier but also uniquely identifies the user. Receiver signals are transmitted to the server through the HTTP protocol, which aggregates and analyzes all collected data. The schematic depiction of this solution is illustrated in Figure 5.

Subsequently, the server conducts an analysis of the received data on a two-dimensional plane. This implies that the plane utilized for localization is partitioned into squares, each possessing a side length denoted as *a* (with a default assumed value of a=1 m). This grid is filled with binary values, zeros and ones, to mimic the spatial distribution of the receiver’s signal coverage. This procedural step is elucidated in Figure 6a,b.

To find the localization of the square where the mobile device is located, the binary values in all squares on the grid are summed arithmetically. An illustrative outcome of this process is presented in Figure 6c. Subsequently, squares with the highest summed values across the entire two-dimensional space are identified. This yields a “probability area” where the localized object (beacon) is highly likely to be located. In scenarios where only one receiver detects the beacon, this area corresponds to the surface area of the receiver’s antenna range, resulting in a uniformly distributed probability with low values. However, as the number of receivers increases, the number of squares with high values also increases, thereby enhancing the probability of accurately localizing the mobile phone. For example, when the antenna coverage of receiver *A* intersects with that of receiver *B*, it suggests a higher likelihood of the beacon being situated within the overlapped region denoted as area *C*, as depicted in Figure 6c.

The accuracy of mobile phone localization can be further improved by simulating multiple beacons concurrently on a single device, a feature supported by the majority of mobile devices. Each of these simulated beacons can emit signals with different transmission power, thereby changing their respective beacon ranges. The distinct ranges imply that individual beacons will fall within the coverage of different receivers, thereby augmenting localization precision.

Differences among the beacons generated on a single device entail alterations to the UUID, ensuring each beacon possesses a unique identifier, and adjusting the actual transmission power (not TX_POWER). In such instances, the schematic representation shifts from that depicted in Figure 5 to that delineated in Figure 7.

In this scenario, the mobile device *M* simulates two distinct beacons: T_SR_, characterized by a short range, and T_LR_, characterized by a long range. Localization of these beacons can be conducted independently. Receiver R_1_ falls within the coverage area of beacon T_SR_, indicating the mobile device’s presence within the range of receiver R_1_, delineated in blue in Figure 7. Beacon T_LR_ has a broader range, encompassing receivers R_1_, R_2_, and R_3_ within its coverage area. Consequently, beacon T_LR_ is likely located within the grid squares assigned a value of 3, highlighted in violet in Figure 7, as the mobile device falls within the range of three receivers: R_1_ (blue), R_2_ (green), and R_3_ (red).

Another aspect worthy of consideration within the algorithm pertains to the frequency of messages transmitted from receivers to the server. Given the distributed nature of the system, although messages are dispatched to the server at nearly uniform intervals denoted by *f*, it is imprudent to presume that precisely one message from each receiver will be received at the same time instance *t* due to inherent imperfections in time synchronization among receivers. Consequently, we aggregate responses from receivers over a predefined temporal interval Δt. Subsequently, for each Δt, a new spatial grid for location determination is constructed. Notably, multiple measurements may be conducted by the same receiver within the temporal window Δt. In the system, scanning is performed every second, although a longer scanning period yields more accurate results. The grid is therefore updated every second. Samples acquired by the system during this time are included in the probability calculation according to the method presented in Figure 6. Each received notification is individually incorporated into the grid, thereby implying that an increased frequency of notifications within Δt augments the likelihood of accurately pinpointing the beacon’s location in proximity to the reporting receiver. Nonetheless, in instances where the beacon is positioned at a considerable distance, resulting in diminished signal strength, messages are received by the receiver with decreased frequency, thereby diminishing the significance of such less reliable devices relative to those situated within closer proximity and thus exhibiting more stable signal reception.

The methodology for determining the localization of a mobile device relies on the aggregation of probability areas generated by multiple simulated beacons. This entails the summation of the respective probability areas pertaining to each aforementioned beacon, followed by the identification of the grid squares exhibiting maximal values. Subsequently, the centroid of the resultant geometric shape is computed, see Figure 8.

An important aspect to consider is the handling of lost data packets. Missing data, whether occurring at the acquisition stage or during reporting, are simply excluded from the algorithm at the given time step. However, in the event of an unfulfilled request from the central system, the missing data are incorporated during the next query, resulting in at most temporary measurement disturbances. Such disturbances are relevant only in scenarios involving rapidly moving objects. It should be emphasized that this situation is relatively unlikely, as the communication is inherently carried out via a fixed infrastructure.

## 5. Measurement Results

We conducted two experiments to investigate system performance and accuracy. The first experiment involved a static setting, covering a small-scale area. Subsequently, the second experiment was executed in a dynamic setting within a large sports hall measuring 45×30 m, aiming to emulate conditions more closely aligned with real-world scenarios. Detailed descriptions of both experiments are provided below.

### 5.1. Experiment 1—Static Measurements

In the initial static experiment, we positioned receivers at the corners of a square grid measuring 10×10 m and recorded measurements of the transmitter’s position at intervals of 1 m (see Figure 9). A total of 121 sets of static measurements were collected, corresponding to an 11×11 grid. We assessed the system’s accuracy in this context by measuring the Euclidean distance between the point determined by the algorithm and the actual object location. We compared our results of static experiment with the HIL algorithm published in [16]. Approximately 25% of the samples exhibit errors not exceeding 1.5 m, while the maximum error for all samples is 5.2 m. The average error is 1.83 m, with a median of 1.73 m and a mode of 2.35 m. Error distribution along the X and Y axes is depicted in Figure 10. Corresponding numerical values are provided in Table 1. Although the null hypothesis regarding the normality of error distributions along the X and Y axes could not be confirmed through the Kolmogorov–Smirnov test, the clustering of the majority of samples around zero error in each dimension supports the efficacy of the centroid as a reliable estimator for mobile device localization.

### 5.2. Experiment 2—Dynamic Measurements

The second experiment aimed to examine the dynamic localization accuracy. We conducted it within the confines of a sports hall of the size 45×30 m, albeit with additional seating areas, which served to mitigate the impact of reflections. We explored various configurations involving alterations to the sampling frequency and the quantity of beacons identifying the device. We observed optimal outcomes when employing a sampling frequency of 1 Hz and utilizing four beacons (transmitted by a single mobile device) with estimated ranges of 5 m, 10 m, 15 m, and 20 m, respectively, under corresponding conditions.

The experiment entailed an individual carrying a mobile device moving along a predefined trajectory within the hall. We recorded his motion via camera, providing a reference for comparing the system’s determined positions.

In this study, we conducted a comparative analysis of our dynamic experimental outcomes with those obtained from the MLoc algorithm as reported in [3]. Direct comparisons between various algorithms for BLE-based localization are challenging due to the influence of numerous variables, including, but not limited to, the number and type of beacons, their localization, sampling frequency, measurement setup, calibration procedures, processing hardware, and others. As such, direct comparisons between studies conducted in different environments should be avoided, as results are often context-dependent. Most studies are conducted in controlled or simulated environments where fingerprinting is typically employed under conditions similar to the measurements, though this is rarely feasible in practical scenarios outside the laboratory. It is crucial to note that a direct comparison between these algorithms is intricate due to variations in the experimental conditions under which each algorithm was originally evaluated. Consequently, the outcomes cannot be directly transposed to our experimental framework, potentially resulting in disparities from the original findings.

Our analysis revealed that, with a signal sampling interval of 1 s, the mean measurement error was 4.1 m for 80% of the samples, while the maximum observed error reached 6.8 m. The system’s performance over a specified time interval, denoted as Δt, is illustrated in Figure 11. Figure 12 presents both the reference trajectory and the actual trajectory of the moving object. The distribution of Euclidean errors is depicted as a histogram in Figure 13, whereas Figure 14 shows the corresponding cumulative distribution function (CDF), in comparison with the static variant of our algorithm as well as the approaches reported in [3,16]. The numerical results associated with these findings are summarized in Table 1.

The reduced localization accuracy in dynamic scenarios is a typical limitation of RSSI-based BLE systems, mainly due to the infrequent measurement of RSSI relative to the velocity of the object. In stationary conditions, multiple measurements can be taken, and averaging over a series of samples provides a robust estimate of the true position, given the Gaussian distribution of RSSI errors. In contrast, dynamic scenarios suffer due to the relatively low sampling rate, typically below 1 Hz, compared to the velocity of a moving object (which can exceed 1.5–2 m/s). Consequently, localization accuracy diminishes as fewer samples are obtained per unit of movement, hindering the ability to calculate reliable averages despite the use of techniques such as Kalman filtering. This performance degradation is a characteristic of most RSSI-based algorithms. However, when the object moves at a sufficiently low speed (e.g., 1.5 m/s in our case), the results remain acceptable within the intended application range. It is also worth noting that the measurements were conducted using widely available, low-cost devices, which were not optimized for precision, further underscoring the practical advantages of the proposed solution. Additionally, dynamic localization accuracy is influenced by several factors not encountered in static measurements, such as the orientation of the smartphone relative to the beacons, antenna characteristics, and interactions with other moving individuals within the experimental area. Unlike most studies, we do not assume a static environment, which enhances the applicability of our approach to real-world scenarios, in contrast to fingerprinting-based methods, which typically require a controlled setting.

## 6. Conclusions and Future Work

Our study presents a novel physical indoor localization system leveraging Bluetooth Low Energy (BLE) beacon technology, with emphasis on its development and evaluation. Static measurements conducted empirically reveal an average error of 1.83 m, a median error of 1.73 m, and a mode error of 2.35 m. Additionally, dynamic localization of a mobile device demonstrates a measurement error of 3.6 m for 70% of samples and 4.6 m for 94% of samples, showcasing superior performance compared to established algorithms. Based on these findings, it is inferred that the system is a dependable tool for localization within expansive structures such as conference halls, exhibition spaces, shopping complexes, or healthcare facilities. The outcomes of this research will be deployed as an operational system within a large-scale exhibition hall in a real-world setting.

Future research endeavors should prioritize the investigation of the transmitting and receiving antenna patterns associated with BLE beacon signals. Furthermore, enhancing the system’s capabilities could involve integrating data collected by the system with various sensors found in mobile devices, including accelerometers, gyroscopes, magnetometers, light sensors, and cameras, employing factor graph data fusion techniques for improved performance.

## Figures and Tables

**Figure 1 sensors-25-05635-f001:**
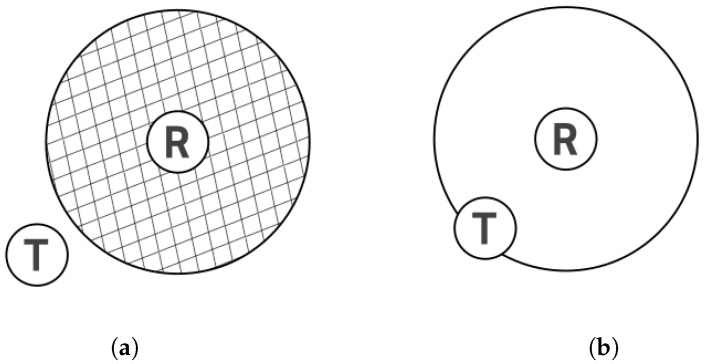
(**a**) T—transmitter, located outside the range of antenna R (receiver). (**b**) T is within the range of R. In this case, it is possible to determine the area where T is located.

**Figure 2 sensors-25-05635-f002:**
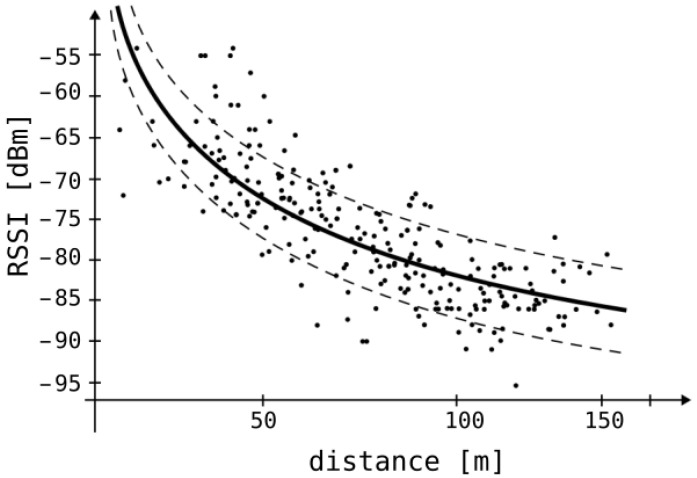
The relationship between the received signal and the distance of the localized device from the transmitter [29].

**Figure 3 sensors-25-05635-f003:**
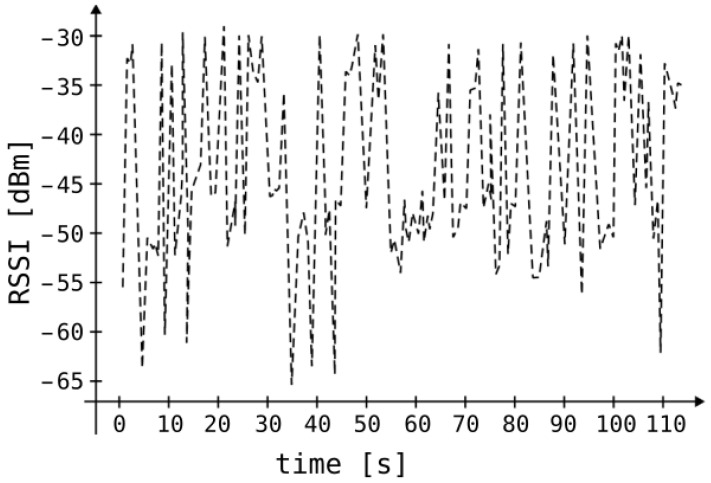
The chart illustrates typical measured changes in RSSI between stationary devices located approximately 7 m apart.

**Figure 4 sensors-25-05635-f004:**
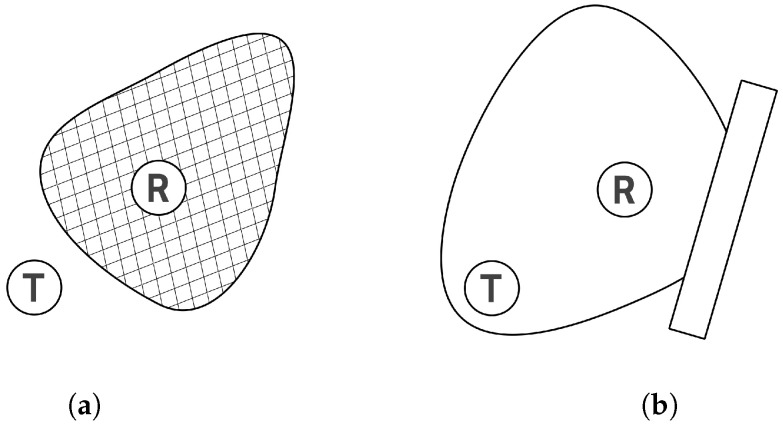
The relationship between range and antenna pattern. (**a**) Directional antenna; T is at the same distance from R as in Figure 1a, but it is out of the receiver’s range. (**b**) Antenna pattern altered by reflection; T is further from R than in Figure 1a, but it remains within the receiver’s range.

**Figure 5 sensors-25-05635-f005:**
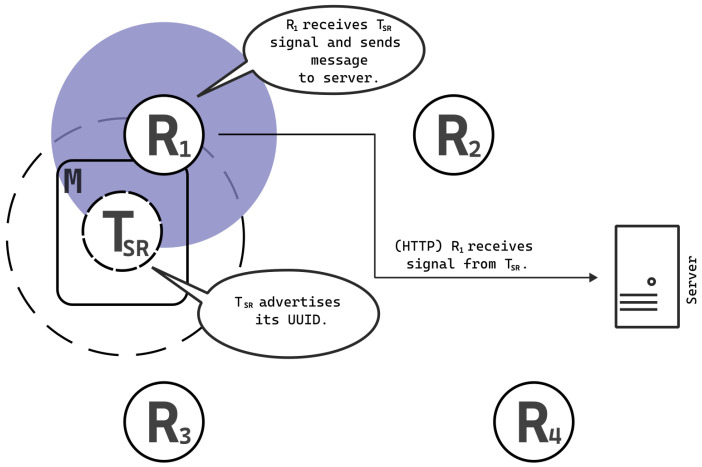
A user utilizes a mobile device to function as a beacon, transmitting packets for advertising purposes. Upon detecting the beacon signal, receivers communicate with the server through the HTTP protocol to relay the information. In this scenario, the transmitter operates at low power, limiting its range to reach only R_1_. The gray, transparent circle around receiver R_1_ represents the estimated antenna pattern, while the dashed circle around transmitter T_SR_ indicates its range.

**Figure 6 sensors-25-05635-f006:**
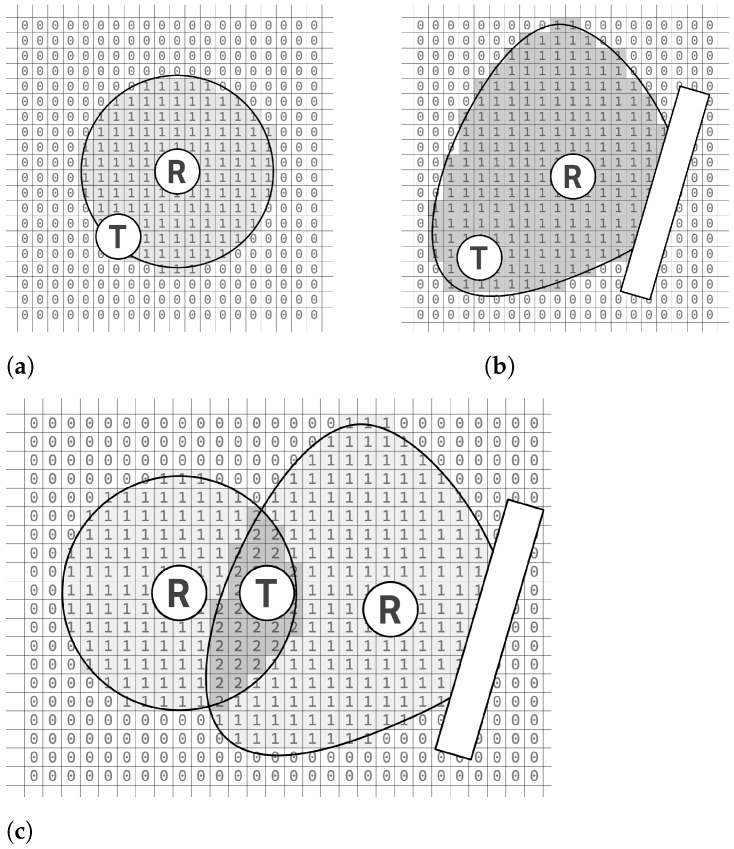
(**a**) Antenna pattern presented on grid. (**b**) Antenna pattern from Figure 4a presented on grid. (**c**) Method of combining patterns to determine the area where the located object is located on grid.

**Figure 7 sensors-25-05635-f007:**
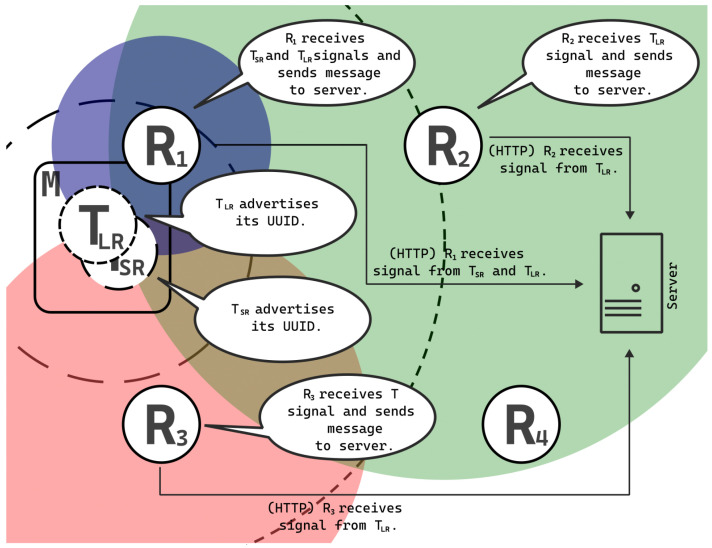
Diagram of a functioning system for two active beacons with different signal transmission powers. A user with a mobile device acting as multiple beacons advertising packets. Stationary receivers, upon hearing the beacon signals, notify the server via the HTTP protocol.

**Figure 8 sensors-25-05635-f008:**
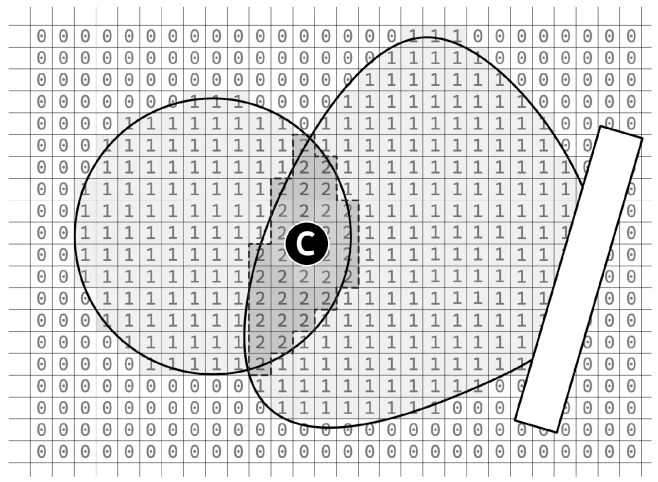
Determining the position of an object (in the form of coordinates x and y) based on the determination of the center of gravity of the area where square values are highest. This area is surrounded by a black dashed line, and the center of gravity is marked with a black circle and labeled as C.

**Figure 9 sensors-25-05635-f009:**
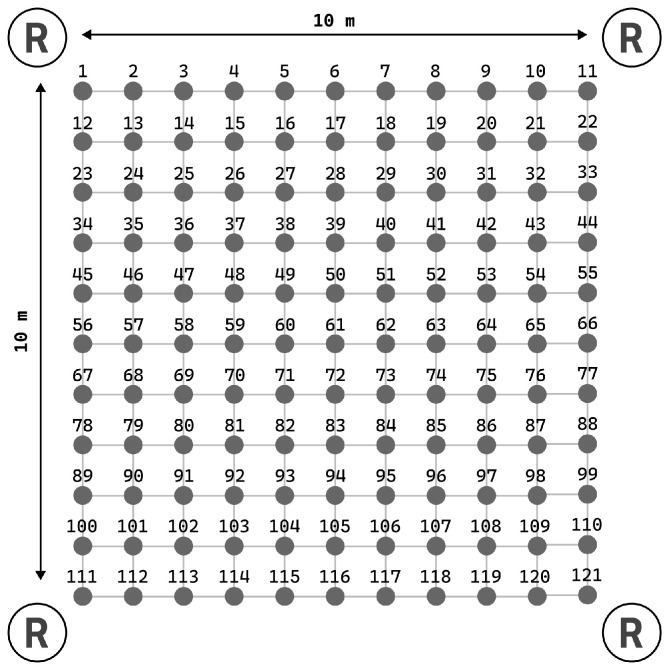
The arrangement of transmitters (gray dots) at measurement points and receivers on a square (1×1 m) grid with dimensions of 10×10 m.

**Figure 10 sensors-25-05635-f010:**
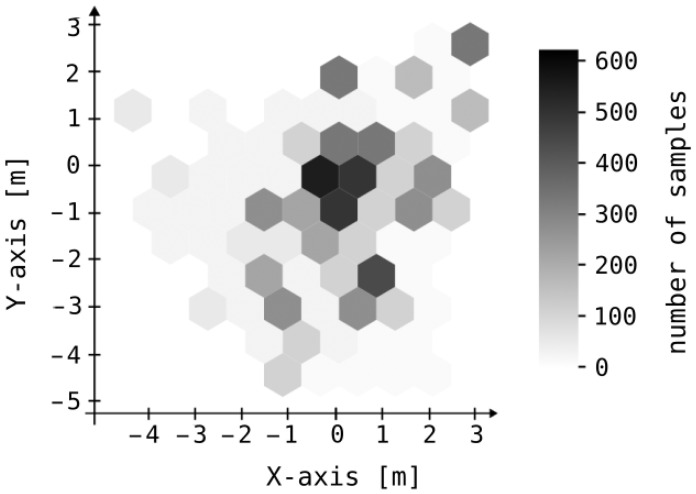
Distribution of position error in the X and Y axes.

**Figure 11 sensors-25-05635-f011:**
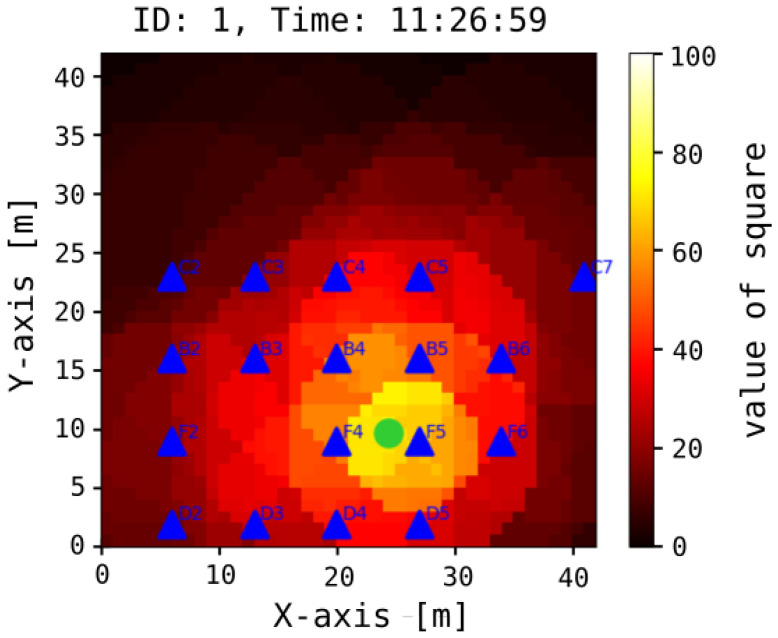
The outcome of the operational system at time *t* for device ID=1 is depicted. Darker areas indicate a lower probability of the object’s presence, while lighter squares suggest a higher likelihood of object occurrence. The green dot represents the centroid of squares with the highest probability. Additionally, blue triangles denote receivers that received signals from any of the transmitters identifying the object.

**Figure 12 sensors-25-05635-f012:**
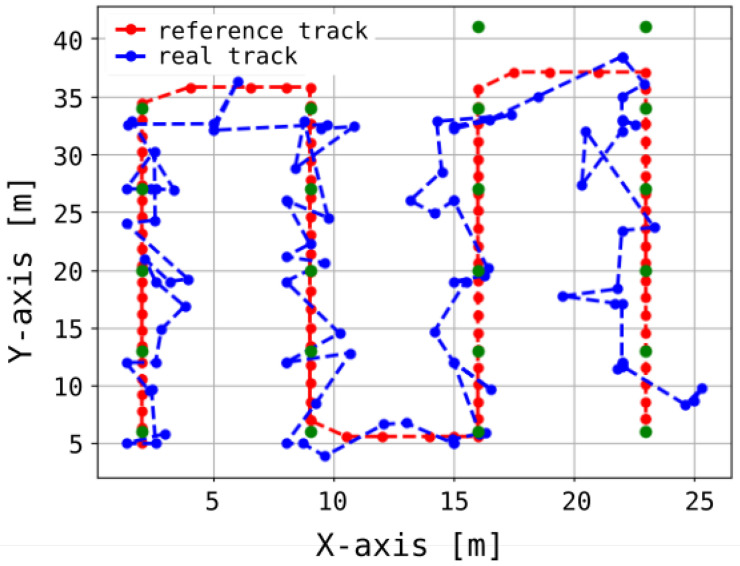
Reference track (red line) and real track (blue line) during the dynamic experiment. Green dots indicate anchor positions.

**Figure 13 sensors-25-05635-f013:**
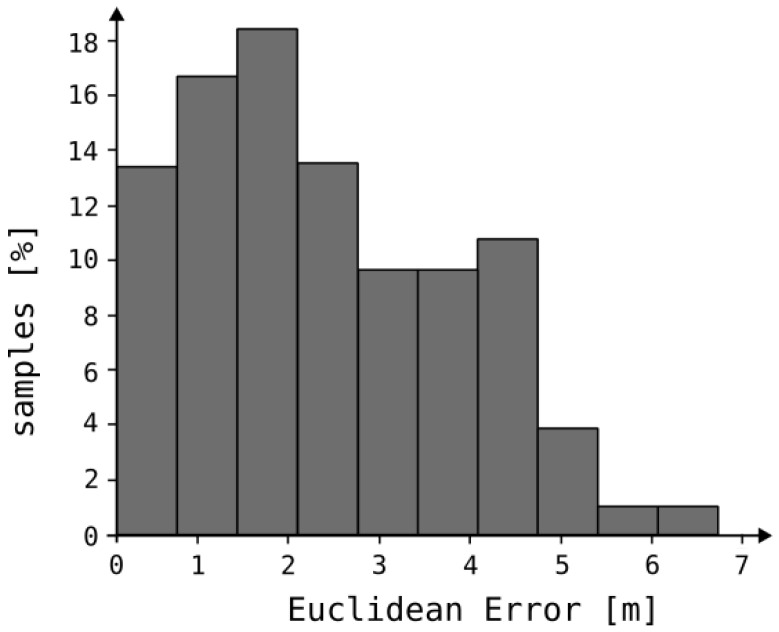
Histogram of Euclidean error.

**Figure 14 sensors-25-05635-f014:**
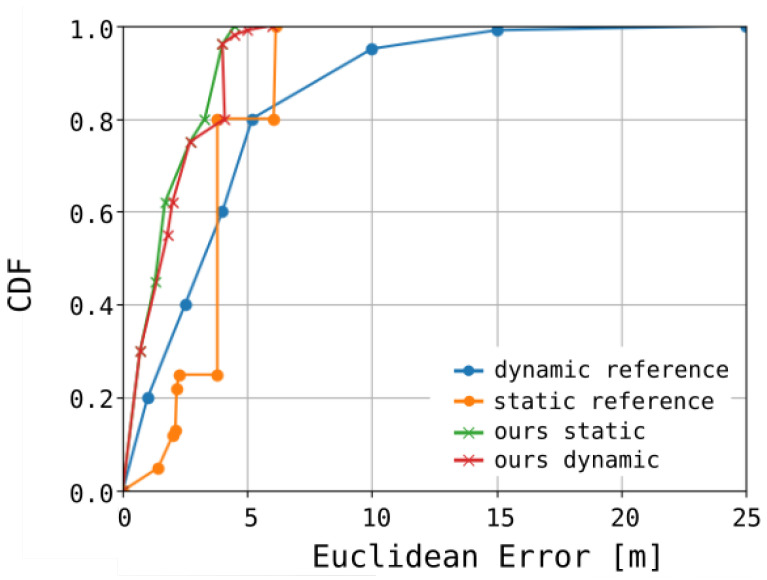
Cumulative distribution function (CDF) of the Euclidean localization error, comparing the results obtained from the static approach reported in [16], the dynamic approach presented in [3], and both static and dynamic variants of the proposed algorithm.

**Table 1 sensors-25-05635-t001:** Comparison of BLE-based indoor localization methods.

Experiment Type	Article	Method	Measurement Grid Density [m^2^]	Best Accuracy for 80% Samples [m]	Median [m]
Static (simulation)	[16]	HIL (RSSI-based BLE indoor localization)	0.05×0.05	3.8	3.1
Static	our algorithm	BLE Probability Grid	1×1	3.3	1.73
Dynamic	[3]	Fingerprinting on BLE + Geomagnetic Field	1×1	5	2.4
	our algorithm	BLE Probability Grid	1×1	4.1	2.21

## Data Availability

We encourage readers to download the source data at https://tulodz-my.sharepoint.com/:u:/g/personal/barbara_morawska_dokt_p_lodz_pl/EYJ1izd7eCxCtwoeGFeoS-0B5OhJaDsmbGkHNtl5fosPXA?e=riwLns (accessed on 9 July 2025).

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
