# Peer review of "Probabilistic Grid System for Indoor Mobile Localization Using Multi-Power Bluetooth Beacon Emulator"

_sensors, 2025, doi:10.3390/s25185635_

Round 1
Reviewer 1 Report
Comments and Suggestions for Authors
In contrast to the conventional RSSI-based methods, the study presents a method for indoor localization based on signal coverage provided by multi-power Bluetooth beacon. This simple and affordable solution might have a significant influence on large-scale implementations such as smart buildings or trade shows.
The main strength of the paper lies in its practical relevance, as the proposed solution is validated through a real implementation and evaluated using a physical testbed.
However, a key weakness is the limited precision achieved, especially when compared to other state-of-the-art solutions that demonstrate higher accuracy.The results of the static evaluation are satisfactory, but the dynamic results seem weaker and need further explanation, especially when it comes to the use of four simulated beacons, their power level settings, and how they relate to estimated distance measurements. Furthermore, comparing the estimated and actual trajectories would make it easier to see how well the system tracks.
The references are acceptable, the related work section should be expanded.
The reference [2] is incomplete.
Author Response
Thank you for very profound and helpful review. Below, we provide our responses to your remarks.
However, a key weakness is the limited precision achieved, especially when compared to other state-of-the-art solutions that demonstrate higher accuracy.
In Section 5.2 – “Experiment 2 - Dynamic measurements” we have added the following explanation:
Direct comparisons between various algorithms for BLE-based localization are challenging due to the influence of numerous variables, including, but not limited to, the number and type of beacons, their localization, sampling frequency, measurement setup, calibration procedures, processing hardware, and others. As such, direct comparisons between studies conducted in different environments should be avoided, as results are often context-dependent. Most studies are conducted in controlled or simulated environments where fingerprinting is typically employed under conditions similar to the measurements, though this is rarely feasible in practical scenarios outside the laboratory.
In Section 2 – “Related work” we have added the following explanation:
Our work distinguishes itself by proposing a practical, hardware-agnostic BLE localization system capable of operating in real-world environments without the need for prior calibration. By integrating a grid-based spatial masking approach with noise-tolerant processing of live RSSI data, our method achieves reliable position estimation using only standard BLE devices. This allows for immediate deployment in dynamic environments where calibration, specialized hardware, or pre-scanning are impractical or resource-intensive.
The results of the static evaluation are satisfactory, but the dynamic results seem weaker and need further explanation, especially when it comes to the use of four simulated beacons, their power level settings, and how they relate to estimated distance measurements.
In Section 5 – “Measurement Results” we have added the following explanation:
The reduced localization accuracy in dynamic scenarios is a typical limitation of RSSI-based BLE systems, mainly due to the infrequent measurement of RSSI relative to the velocity of the object. In stationary conditions, multiple measurements can be taken, and averaging over a series of samples provides a robust estimate of the true position, given the Gaussian distribution of RSSI errors. In contrast, dynamic scenarios suffer due to the relatively low sampling rate, typically below 1Hz, compared to the velocity of a moving object (which can exceed 1.5–2m/s). Consequently, localization accuracy diminishes as fewer samples are obtained per unit of movement, hindering the ability to calculate reliable averages despite the use of techniques such as Kalman filtering.
This performance degradation is a characteristic of most RSSI-based algorithms. However, when the object moves at a sufficiently low speed (e.g., 1.5m/s in our case), the results remain acceptable within the intended application range. It is also worth noting that the measurements were conducted using widely available, low-cost devices, which were not optimized for precision, further underscoring the practical advantages of the proposed solution. Additionally, dynamic localization accuracy is influenced by several factors not encountered in static measurements, such as the orientation of the smartphone relative to the beacons, antenna characteristics, and interactions with other moving individuals within the experimental area. Unlike most studies, we do not assume a static environment, which enhances the applicability of our approach to real-world scenarios, in contrast to fingerprinting-based methods, which typically require a controlled setting.
Furthermore, comparing the estimated and actual trajectories would make it easier to see how well the system tracks.
We also appreciate the reviewer’s suggestion to include trajectory comparisons. We have incorporated these, and they are now presented in Figure 12.
The references are acceptable, the related work section should be expanded.
Related Work Section has been significantly improved and substantially expanded to provide a broader context for our contribution. It now includes discussion of RSSI-based BLE localization methods, both low- and high-precision; visibility-based, graph-based localization approaches; direction-finding techniques; multi-carrier phase difference methods integrating BLE with IMU data; as well as fingerprinting-based techniques, including LSTM-enhanced variants. Furthermore, we highlight the distinctions between these methods and our proposed approach, emphasizing its hardware-agnostic design, lack of calibration requirements, and applicability in real-world, dynamic environments.
The reference [2] is incomplete.
We appreciate the careful review. The reference has been amended accordingly.
Reviewer 2 Report
Comments and Suggestions for Authors
This paper presents an enhanced method for Bluetooth-based location sensing. Unlike traditional Bluetooth proximity methods where the device acts as the receiver, the proposed system adopts a reversed approach: receivers are installed in the environment to form a sensing infrastructure. On top of it, the paper also introduces a method that uses multiple transmission power levels to encode additional spatial information in the beacon signals. Experimental results in both static and dynamic settings demonstrate improvements over existing systems.
The system and proposed method in the paper are interesting. The use of multi-transmission-power beacons effectively improves localization performance, especially when the number of receivers is limited. Additionally, I want to highlight that the figures in the manuscript are well-designed and easy to understand.
However, I would like to raise several concerns that the authors may consider to improve the paper:
- The first three sections (Introduction, Background, and Problem in Bluetooth Beacon Indoor Localization) are not well structured. Typically, these components should be integrated into a single Introduction section that provides background, introduces the problem and motivation, and offers an overview of the contributions and method. However, the current structure lacks clarity. Additionally, some descriptions are not relevant. For example, the extensive discussion on Eddystone beacons seems unnecessary, as the proposed method can be applied to any beacon system.
- In the Algorithm section, the paper does not clearly describe how the likelihood is modeled (e.g., binary or frequency-based), how joint probabilities are computed, and how data collection is conducted (e.g., duration at each position).
- The method does not address missing values. Packet loss is common in real-world beacon systems and should be considered as part of the system design.
- The experimental section lacks figures that illustrate the results. Including CDF curves or distributions of likelihood/probability would help readers better understand and evaluate system performance.
Author Response
Thank you for very profound and helpful review. Below, we provide our responses to your remarks.
The first three sections (Introduction, Background, and Problem in Bluetooth Beacon Indoor Localization) are not well structured. Typically, these components should be integrated into a single Introduction section that provides background, introduces the problem and motivation, and offers an overview of the contributions and method. However, the current structure lacks clarity.
The structure of the article has been substantially revised. The “Background” section, previously numbered as Section 2, has been partially merged with the “Introduction” section, while the “Related Work” section has been moved to the second position.
Additionally, some descriptions are not relevant. For example, the extensive discussion on Eddystone beacons seems unnecessary, as the proposed method can be applied to any beacon system.
We acknowledge this point and have removed the discussion on Eddystone beacons, as it was not essential to the central thesis of the paper.
In the Algorithm section, the paper does not clearly describe how the likelihood is modeled (e.g., binary or frequency-based), how joint probabilities are computed, and how data collection is conducted (e.g., duration at each position).
In Section 4 – “Algorithm” we have added the following explanation:
In the system, scanning is performed every second, although a longer scanning period yields more accurate results. The grid is therefore updated every second. Samples acquired by the system during this time are included in the probability calculation according to the method presented in Figure 6.
The method does not address missing values. Packet loss is common in real-world beacon systems and should be considered as part of the system design.
We acknowledge this concern and have included the following explanation in Section 4 – “Algorithm”:
An important aspect to consider is the handling of lost data packets. Missing data, whether occurring at the acquisition stage or during reporting, is simply excluded from the algorithm at the given time step. However, in the event of an unfulfilled request from the central system, the missing data are incorporated during the next query, resulting in at most temporary measurement disturbances. Such disturbances are relevant only in scenarios involving rapidly moving objects. It should be emphasized that this situation is relatively unlikely, as the communication is inherently carried out via a fixed infrastructure.
The experimental section lacks figures that illustrate the results. Including CDF curves or distributions of likelihood/probability would help readers better understand and evaluate system performance.
We appreciate this feedback. To improve clarity, we have added Cumulative Distribution Function (CDF) plots alongside histograms and visual comparisons of the real and reference trajectories. These additions aid in more comprehensively illustrating system performance.
Round 2
Reviewer 2 Report
Comments and Suggestions for Authors
Thanks authors for addressing the comments. Overall I am satisfied with the current version. Some extra comments below.
- The introduction section now is more concise than the previous version, but it still lacks an overview of the proposed approach. The current focus is only on the background and the challenge. It would be easier for reader to follow if the intuition behind the solution and an overview are provided.
- In the experiment, the authors have added a CDF to present the error distribution. My intention in the previous comment is to add a comparison with work 15 and/or comparison between static and dynamic so that reader will have a clear understanding. The CDF of the method alone contains the similar information to the histogram.
- The authors mentioned the application of beacons in COVID. An interesting application the author might refer to is: J. Tan, E. Sumpena, W. Zhuo, Z. Zhao, M. Liu and S. . -H. G. Chan, "IoT Geofencing for COVID-19 Home Quarantine Enforcement," in IEEE Internet of Things Magazine, vol. 3, no. 3, pp. 24-29, September 2020, doi: 10.1109/IOTM.0001.2000097.
Author Response
The introduction section now is more concise than the previous version, but it still lacks an overview of the proposed approach. The current focus is only on the background and the challenge. It would be easier for reader to follow if the intuition behind the solution and an overview are provided.
Thank you for the advice. We have revised the Introduction section by explicitly adding our contribution.
“The main contribution of this paper is the introduction of a novel, cost-effective, BLE-based approach for indoor mobile device localization. Unlike traditional methods that rely on Received Signal Strength Indicator (RSSI), the proposed method leverages spatial signal coverage generated by multipower Bluetooth emulators. By constructing a probability grid derived from receiver range data, the approach significantly enhances localization accuracy. Experimental validation demonstrated its effectiveness outperforming similar solutions. Beyond the experimental setup, the system is currently being deployed at trade fairs to monitor attendee movement and support the optimization of stand distribution.”
In the experiment, the authors have added a CDF to present the error distribution. My intention in the previous comment is to add a comparison with work 15 and/or comparison between static and dynamic so that reader will have a clear understanding. The CDF of the method alone contains the similar information to the histogram.
Thank you for your valuable comment. In Figure 14, we present the CDFs of both the solutions used for comparison (static and dynamic algorithms) as well as the results from the static experiments of our algorithm.
Since a direct comparison of our algorithm with HIL is not straightforward, we converted the values in m² to meters as follows. First, we took the square root of the area, and then we multiplied the result by 10/7, which corresponds to the ratio between the distance between our beacons and the distance used in the algorithm under comparison.
Additionally, the publication previously listed as reference 15 is now numbered as reference 16, following the inclusion of the publication suggested by the Reviewer.
The authors mentioned the application of beacons in COVID. An interesting application the author might refer to is: J. Tan, E. Sumpena, W. Zhuo, Z. Zhao, M. Liu and S. . -H. G. Chan, "IoT Geofencing for COVID-19 Home Quarantine Enforcement," in IEEE Internet of Things Magazine, vol. 3, no. 3, pp. 24-29, September 2020, doi: 10.1109/IOTM.0001.2000097.
Thank you for highlighting this valuable publication (now listed as reference 8). We have incorporated it into the Introduction section, specifically within the discussion of COVID-19-related systems.